# Advances in Electrochemical and Acoustic Aptamer-Based Biosensors and Immunosensors in Diagnostics of Leukemia

**DOI:** 10.3390/bios11060177

**Published:** 2021-05-31

**Authors:** Tibor Hianik

**Affiliations:** Faculty of Mathematics, Physics and Informatics, Comenius University, Mlynska Dolina F1, 842 48 Bratislava, Slovakia; tibor.hianik@fmph.uniba.sk; Tel.: +421-2-602-95-683

**Keywords:** leukemia, biosensor, nucleic acid aptamers, immunosensors, electrochemical detection, acoustic sensors

## Abstract

Early diagnostics of leukemia is crucial for successful therapy of this disease. Therefore, development of rapid, sensitive, and easy-to-use methods for detection of this disease is of increased interest. Biosensor technology is challenged for this purpose. This review includes a brief description of the methods used in current clinical diagnostics of leukemia and provides recent achievements in sensor technology based on immuno- and DNA aptamer-based electrochemical and acoustic biosensors. The comparative analysis of immuno- and aptamer-based sensors shows a significant advantage of DNA aptasensors over immunosensors in the detection of cancer cells. The acoustic technique is of comparable sensitivity with those based on electrochemical methods; moreover, it is label-free and provides straightforward evaluation of the signal. Several examples of sensor development are provided and discussed.

## 1. Introduction

Human blood is a unique substance consisting of white blood cells, red blood cells, and platelets. The white blood cells are responsible for fighting infection. A disease characterized by uncontrolled proliferation of white blood cells is leukemia. Leukemia is associated with the body’s blood-forming tissues, including the bone marrow and the lymphatic system. According to the analysis of World Health Organization (WHO) cancer databases, leukemia incidence varies considerably by geography and subtype.

Generally, there are several types of leukemia, of which acute lymphoblastic leukemia (ALL) is among the most dangerous. The incidence of ALL is most common in children but rarely affects adults [1]. The most recent data on the population-based occurrence of leukemia and comparison with other types of cancers, including incidence and mortality, were compiled in the recent publication by Siegel et al. [2].

ALL is the most dangerous cancer attacking children from 2 to 5 years old, and it accounts for 73% of all cases of leukemia in children. The 5-year survival rate of ALL increased from 43% in the mid-1970s to 87% in 2005 [3]. In the majority of cases of childhood leukemia, the reason is unknown. Presently, there is not a full understanding of the exact mechanisms of leukemia development. It seems that it occurs from a combination of genetic and environmental factors.

Therefore, there is an urgent need for the development of effective diagnostic and therapeutic strategies. Early diagnosis is crucial for the survival and positive prognosis of these cancer patients.

The appropriate diagnosis and classification of leukemias in clinical practice rely on the simultaneous application of multiple techniques. In order to achieve a complete diagnosis in a patient’s specimen, it is necessary for a modern clinical laboratory to be equipped with sophisticated novel analytical tools that are operated by skilled and experienced personnel. Currently, the conventional techniques of cytomorphology and histomorphology are combined with cytochemistry and multicolor flow cytometry (FACS). Flow cytometry is a widely used technique in routine hematology for analysis of specific markers mostly present on the cell surface of different cell types in a heterogeneous cell population. Furthermore, the mentioned diagnostic techniques are supplemented by fluorescence in situ hybridization (FISH) and polymerase chain reaction (PCR)-related methods. FISH represents a molecular cytogenetic technique utilizing fluorescent probes that bind specific complementary DNA sequences on chromosomes, which allows visualization and mapping of the genetic material in a given individual cell within the population of leukemia cells [4]. PCR is a widely used amplification method, which allows preparation of DNA in enough quantity for detailed analysis. The advanced PCR, quantitative reverse transcriptase PCR (QRT-PCR), allows detection of the fusion and mutated genes, and it is being employed for detection of minimal residual disease (MDR) in certain types of leukemia [5]. Another variant, the droplet digital PCR (ddPCR), is under investigation [6]. There are also attempts to increase the sensitivity for detection of small clones within MRD [7]. During the last two decades the diagnostic accuracy and efficiency of the above-mentioned methods were significantly improved by application of microarrays for gene expression profiling, which can predict all relevant sub-entities of leukemia. The microarrays approach allows the simultaneous detection of the expression of nearly all human genes in one assay. Thus, it provides very detailed insight into gene regulation and related pathological alterations on the transcriptional level present in tested leukemia cells [8]. Further progress in diagnostic methods resulted in development of next-generation sequencing (NGS), also known as high-throughput sequencing. NGS allows sequencing of DNA and RNA more quickly than previous technology. NGS allows very prompt identification of rare cancer mutations, translocations, inversions, insertions, and deletions in clinical samples. This provides the molecular rationale for an individual patient-tailored therapeutic and prognostic approach [9]. In order to improve personalized medicine for cancer patients, the integration of several sophisticated technologies such as genomics, transcriptomics, proteomics and metabolomics is employed to provide a more comprehensive view of leukemogenesis [10].

However, current trends in personalized medicine demand development of fast and sensitive tests that can be used for preliminary screening in the doctor’s consulting room, at the patient bed side, or even at home. For this purpose, the development of biosensors can be a rather useful solution. Biosensors are devices composed of a receptor that recognizes the analyte of interest (for example, glucose, cholesterol, cancer marker, etc.), a transducer that transduces the binding event at the sensing surface into the measured signal (electrical, optical, or acoustic). Finally, the analyzer evaluates the concentration of the analyte in the sample [11]. So far, the most successful application of biosensors has been in the field of endocrinology. Biosensors are widely used for monitoring blood glucose in diabetic patients at clinics as well as at home. The successful application of biosensors in medicine has also focused on detection of cholesterol and neurotransmitters [12].

The accuracy and efficiency of cancer diagnostics may be improved by using specific markers. For example, as a potential cancer marker for T-cell acute lymphoblastic leukemia (T-ALL), the protein tyrosine kinase 7 (PTK7) has been identified. It is a transmembrane receptor protein that was discovered in a series of leukemia cell lines. PTK7 is overexpressed on CCRF-CEM and Jurkat cells and at other tumors such as colon and gastric cancer. For biosensor development it is also important to select the recognition element. The monoclonal antibodies were so far mostly used for this purpose. The immunosensors are typically characterized by good sensitivity and selectivity. However, the drawback consists in impossibility of immunosensor regeneration. Therefore, only disposable immunosensors can be used. Additional drawbacks of immunosensors are low stability and high costs of monoclonal antibodies [13]. As an alternative, the nucleic acid aptamers can be used as receptors. Aptamers are DNA or RNA oligonucleotides that are composed of typically 15–80 nucleotides that in water solution fold into 3D conformation forming a specific binding site to the analyte of interest. They are specially selected by the method SELEX (systematic evolution of ligands by exponential enrichment), developed in 1990 independently by three laboratories in the USA [14]. The advantage of aptamers is high affinity to the target (the constant of dissociation, K_D_, is typically in the level of 1–100 nM) [15]. Aptamers are also more stable in comparison with antibodies and can be selected for almost unlimited types of analyte, including small molecules, proteins, viruses, bacteria, or cells. Another important issue in the development of a biosensor is the selection of the proper chemical modification of the receptors. In this respect, nucleic acid aptamers are ideal because they allow various chemical modifications that also improve their stability. It is also important to provide antifouling properties of biosensors for their application in complex liquids such as blood or blood plasma [16].

Several nucleic acid aptamers were developed so far for application in cancer diagnostics [17]. For example, DNA aptamer sgc8c specific to PTK7 has been selected by Pan et al. [18] and used as a recognition element in various biosensors, also called aptasensors. Aptasensors for cancer diagnostics described in this review are based on electrochemical and piezoelectric transducers. Electrochemical biosensors require simple instrumentation and offer high specificity. Aptamers are immobilized in direct contact with an electrode surface, and changes in electrical values (potential, current, impedance, and conductance) are measured upon binding to the analyte. Piezoelectric sensors use quartz crystals, which resonate under the application of an external high frequency voltage. The binding of the analyte to the aptamers immobilized on the surface of quartz crystal resulted in resonant frequency shift due to mass changes. Piezoelectric aptasensors are advantageous for clinical diagnostics, due to the possibility of real-time detection.

In this review the current state of the art in the development of immuno- and aptasensors based on electrochemical and acoustic principles is presented, with focus on the comparison of the efficiency of antibodies and nucleic acid aptamers. It is demonstrated that aptamer-based technology has opened new opportunities for efficient diagnostics of leukemia.

## 2. Electrochemical and Acoustic Immunosensors

### 2.1. Antibodies and the Methods of Their Immobilization at Surfaces

Immunosensors are based on the specific interaction between an antibody immobilized at the sensor surface and an antigen (or analyte) of unknown concentration. The antibody forms a strong immunocomplex with the antigen and thus produces a physicochemical response that is transformed by the corresponding transducer into the measured physical signal. The main requirements for biosensors, including immunosensors, are (1) low limit of detection (LOD), which should typically be in the range of pM–nM; (2) high specificity to the analyte; (3) sufficiently wide dynamic range (range of concentrations in which response of the sensor for various concentrations of the analyte is close to the linear); (4) fast response (typically in the range of seconds); (5) anti-fouling properties (possibility to operate in complex biological liquids and providing minimal interferences with other compounds); (6) possibility of regeneration (multiple use of the biosensor); (7) sufficiently high stability (possibility of storage and transportation to the patient bed site or for point of care diagnostics in personalized medicine); (8) easy to use and preferably automatic analysis of the sample; and (9) low cost compared to currently used clinical methods. The crucial steps in immunosensor preparation consist of (a) optimal immobilization procedure allowing proper orientation of binding site of antibodies; (b) optimal surface density of antibodies; (c) large surface area; and (d) selection of optimal matrix for antibody immobilization (for example nanostructures). Very important also is production of specific antibodies (for example, monoclonal antibodies (mAbs)) such as those for cancer diagnostics and immunotherapy [19]. The antibodies (Abs) are glycoproteins that belong to the immunoglobulins (Ig). They have a Y-shaped structure, schematically presented in Figure 1. The Ab is composed of two identical pairs of peptide chains—light and heavy—that are stabilized by disulfide bonds. Ab is composed of two Fab and one Fc regions. The Fc part is responsible for immunoglobulin subclass (IgA, IgD, IgE, IgG, and IgM), while Fabs recognize the antigen. Each Fab contains two constant domains for heavy (CH) and light (CL) chains as well as two variable parts for heavy (VH) and light (VC) chains, respectively. The variable domains are controlled genetically and are responsible for the generation of Abs for practically unlimited antigens [20,21]. Polyclonal antibodies (pABs) can be produced by immunized animals (rat, rabbits, goats, or sheep). pABs are frequently used in immunosensors; however, their specificity is relatively low because they represent a pool of Abs secreted by B cells. The IgG is the main Ab found in the blood [22]. The monoclonal antibodies are of special interest because they provide high specificity to the antigen. They are produced by molecular biology and biotechnology methods using specific B cell clones. Current biotechnology is also focused on the preparation of single chain antibody fragments (csAb) that can be used not only as recognition elements in immunosensors but also have increased application in cancer immunotherapy due to improved pharmacokinetics and tissue penetration [23,24]. New trends in the development of receptors for immunosensors involve application of genetic engineering for the production of mono- or multivalent fragments of recognition sites of Abs. These fragments have similar recognition properties like Abs; however, they can be produced more economically using recombinant expression methods [25].

The crucial step in immunosensor preparation consists of the immobilization of antibodies at the solid support in order to provide proper orientations of Fab fragments and thus maximize the number of binding sites for the antigens. Another important parameter is the optimal density of Abs. Sufficiently high density of receptors not only provide higher sensitivity but also minimize undesirable interference of the analyte with the naked biosensor surface [26,27]. At the same time immobilization of antibodies using special linkers that provide antifouling surface properties is among the current trends in biosensor development [16]. The simplest method of Abs immobilization is physical adsorption (Figure 2A).

The solid support consisting of various materials (gold, carbon, graphene, conducting polymers, etc.) is incubated with Abs. Due to electrostatic, hydrophobic, or van der Waals interaction, the Abs are adsorbed at the surface, but both Fab and Fc parts lie flat. This results in restricted access of the antigen to Abs [28]. Such a construct is simple and low cost; however, it has several disadvantages such as: (1) Abs are randomly oriented and some of the binding sites (Fab) are blocked; (2) certain part of Abs can be denatured; (3) Abs can be replaced by other molecules in the sample; (4) washing of the surface can cause removal of certain portion of Abs; and (5) low reproducibility and impossibility of surface regeneration. This method is, therefore, not suitable for practical applications.

**Covalent immobilization**. This type of immobilization is most effective with respect to the stability of the immunosensor. Depending on the surface (gold, silver, carbon, graphene, etc.), various conjugation chemistry can be used. Among most popular are those based on carbodiimide chemistry. In this method, the carboxylic groups of the molecules (for example 11-mercaptoundecanoic acid (MUA) chemisorbed at gold surface) are activated by *N*-(3-dimetylaminopropyl)-*N*′-etylcarbodiimide (EDC), *N*-hydroxisuccinimide (NHS). Addition of Abs results in covalent binding of free NH_2_ groups to the activated COO^−^ residues (Figure 2B). Alternatively, the carboxyl groups at the Ab are activated by EDC/NHS and covalently bind to the amino groups of cysteamine chemisorbed at the gold surface (Figure 2C). Recently silanized long-period fiber gratings (LPFG) coated by polyelectrolytes were also used for covalent immobilization of antibodies using glutaraldehyde crosslinking. This platform has been used for optical detection of the bacterial pathogen *Staphylococcus aureus* [29]. Another method of immobilization is based on the possibility of dissociation of the disulfide bonds of Abs. The resulting Fab molecules can be chemisorbed at a gold. Alternatively, Abs containing terminal SH groups at the cysteine amino acid can be covalently linked to the maleimide molecules adsorbed at the surface (Figure 2D). The advantage of covalent immobilization using NHS/EDC and maleimide chemistry results in a stable immunosensor. However, even in this case the Fabs are randomly oriented and some of them can be blocked for antigen binding. An exception is, however, covalent immobilization of half Fabs fragments, as mentioned above [30].

**Proteins A and G for Abs immobilization.** The possible solution for oriented Abs immobilization is using A or G proteins that specifically bind to Fc fragments. Proteins A and G are localized in the bacterial cell wall of *Staphylococcus aureus* and G *streptococci*, respectively. These proteins can be immobilized at various surfaces by common covalent immobilization techniques. The addition of antibodies results in strong binding between proteins A or G to Fc fragment, providing good orientation of the Ab binding sites [31,32,33,34] (Figure 2E).

**Biotin-modified Abs.** Fc chain modified by biotin can be immobilized at the surface covered by the monolayer composed of avidin, streptavidin, or neutravidin. Due to the strong affinity of biotin to the immobilized molecules a stable Abs monolayer is formed (Figure 2F) [35]. Another advanced method of immobilization of antibodies, including their engineered Fab fragments, was explained in review by Li and Chen [26].

### 2.2. Electrochemical and Acoustic Methods for the Detection Ab–Ag Interactions

#### 2.2.1. Electrochemical Methods

Electrochemical methods such as amperometry, potentiometry, conductometry, and electrochemical impedance spectroscopy (EIS) are rather widespread in immunosensors development [36]. The Abs as well as most of the potential antigens—cancer markers, cancer cells—are electrochemically inactive. Therefore, electrochemical sensors should use special conductive surfaces, such as gold, silver, carbon, glassy carbon, graphene, graphene oxide, carbon nanotubes, conducting polymers, or various hybrid nanomaterials for Abs immobilization. The redox couple, redox labels, metal nanoparticles, or quantum dots should also be used for Ab–Ag detection. In this section, we briefly explain the basic principles of the methods most often used for electrochemical detection of Ab–Ag interactions. More detailed explanation of the principles of various electrochemical methods used in immunosensors can be found in a comprehensive review by Kokkinos et al. [36].

The most frequently used electrochemical strategy for detection of the Ab–Ag interaction is based on the labeling of the Ab by an enzyme, such as horseradish peroxidase (HRP) or alkaline phosphatase (ALP). These are so-called enzyme-based immunosensors. HRP converts its substrate—the hydrogen peroxide, H_2_O_2_—into H_2_O with the release of 2 electrons that at certain potential move to the electrode surface. However, the direct use of HRP in an enzyme sensor requires application of the surfaces with good conductivity and the ability to immobilize higher amount of Abs. This can be, for example, a glassy carbon electrode (GCE) covered by gold nanowires or ZnO nanorods [37] or single-walled carbon nanotubes [38]. If HRP is relatively far from the surface and high sensitivity of detection is required, the redox mediators must be used for increasing the redox current. Rather often, mediators are for example ferrocene carboxylic acid (FCA), N,N,N′,N′-tetramethylbenzidine (TMB) [39], ferro/ferricyanide redox couple, or benzoquinone [11] (Figure 3A). ALP converts the non-electroactive substrate 1-naphtyl phosphate (1-NP) into the electroactive product 1-naphtol (Figure 3B).

The principles of enzyme immunosensors have been used in the most applied enzyme-linked immunosorbent assay (ELISA) [36]. Commercial ELISA kits are available in the market for the detection of various compounds. The sensitivity of ELISA is rather high, with limit of detection (LOD) typically in the range 10^−12^–10^−9^ M. However, in the commercial kits, colorimetry is used for detection of the analyte instead of electrochemistry. A colorless enzyme substrate, for example, p-nitrophenyl phosphate (pNPP), is transformed by alkaline phosphatase into yellow p-nitrophenol [40]. The enzyme reaction can therefore be observed even by naked eye and quantitatively evaluated by absorption spectroscopy at certain wavelengths. The fluorometry based ELISA kits are also available. The advantage of ELISA is its good sensitivity and selectivity. However, this method is multistep in preparation, time consuming, and also requires optical readers. The scheme of an ELISA assay in various formats is presented in Figure 4.

The simplest direct ELISA format is based on the detection of antigens adsorbed at the solid support (Figure 4a). The addition of the enzyme-labeled Ab converts the colorless substrate into a colored compound detectable by colorimetry. In the indirect assay the Abs selective to the antigen of interest are immobilized at the surface. After the addition of the sample containing antigen, the secondary antibody labelled by the enzyme is added. This secondary antibody selectively binds to the primary Abs (Figure 4b). If Abs able to interact with two different binding sites of antigens are available, then the sandwich assay can be used (Figure 4c). Each of the formats presented in Figure 4 can be transformed into the competitive ELISA assay. For example, the analyte containing antibody is mixed with a fixed concentration of the enzyme labelled Abs. Such a mixture is then added at the surface with the immobilized primary antibody, like those shown in Figure 4c. In such a format the colorimetric signal will be inversely proportional to the concentration of antigens [36].

#### 2.2.2. Acoustic Methods

Mass-sensitive quartz crystal microbalance (QCM) is a very efficient method for the study of affinity interactions at surfaces. This method is especially useful for the detection of proteins or cells that have of enough mass (typically more than 2 kDa) for inducing significant changes of resonant frequency. However, the thickness shear mode method (TSM) and QCM with dissipation (QCM-D) also allow analyzing the surface viscosity by measuring motional resistance, R_m_, or dissipation factor (ΔD) [42,43]. These values are sensitive to the viscoelastic properties of the sensing layers. The surface viscosity can also be affected by binding to the surface of low molecular compounds, such as toxins, antibiotics, short peptides, and others.

In contrast with electrochemical methods, the QCM is a single physical probe assay. It is based on the measurement of the properties of a propagated acoustic wave. Such a noninvasive method allows the study of the binding of cells to the QCM transducer surface without affecting the cell properties. However, the interaction of the cells with the receptors immobilized at the crystal surface is a complex process that affects not only the resonant frequency but also includes viscosity contribution. Therefore, the relation between the frequency and mass changes are not straightforward and cannot be described simply by the Sauerbrey Equation (1) [44],
(1)$$\Delta f=\frac{-2f_{0}^{2}n\Delta m}{A\cdot \sqrt{\mu_{q}\rho_{q}}}$$
where *f*_0_ is the fundamental frequency of the piezocrystal (typically 5–20 MHz), *n* is the overtone number, Δ*m* is the mass change, *A* is the active sensor area (typically 0.2 cm^2^), *µ_q_* = 2.95 × 10^11^ dyne/cm^2^ is shear modulus, and *ρ_q_* = 2.648 g/cm^3^ is the density of quartz, respectively. Equation (1) is strongly valid for rigid thin films in a vacuum (Sauerbrey effect). In this case, the theoretical mass sensitivity, *S*, can be determined as: *S* = Δ*m*/Δ*f* = 2*f*_0_^2^/*ρ_q_νn*, where *ν* is the velocity of the propagated acoustic wave. Limit of detection (LOD) can be estimated as LOD = Δ*m_min_* = Δ*f_min_*/*S*, where Δ*f_min_* is the minimum detectable frequency shift [45]. In a liquid, the viscoelastic contribution should be considered (Kanazawa effect) [45,46]:(2)$$\Delta f=\frac{-2f_{0}^{2}\left(m_{c}+m_{L}\right)}{Z_{cq}}\;$$
where *m_c_* is the surface mass density of the coating (Sauerbrey effect), *m_L_* = *ρ_L_**δ_L_*/2 is the equivalent surface mass density of the liquid, which is characterized by an exponential damped profile (Kanazawa effect). *ρ_L_* and *δ_L_* are the water density and the penetration depth of acoustic wave in the water, respectively (Figure 5A). The penetration depth in water can be determined as: *δ_L_* = (2*η_L_*/*ωρ_L_*)^1/2^ where *η_L_* is viscosity, and *ω* = 2*πf* is the circular frequency of oscillations. For the crystal of fundamental frequency, 5 MHz, the *δ_L_* value in water, is 0.25 µm [47]. Due to the viscosity effect, the mass changes, determined from the changes in frequency, are in general overestimated. Therefore, accuracy in the interpretation of the data, especially in the case of the cell detection, is crucial. In order to evaluate the contribution of viscosity into the resonant frequency changes, the determination of motional resistance, *R_m_*, is required. It is possible by fitting the impedance spectra of the piezocrystal by a Butterworth–van Dyke (BvD) equivalent circuit (Figure 5B). In this circuit, *C*_0_ = *εA*/*h* is the parallel electrical capacitance, *L* = 1/*ω*^2^*C* is the motional inductance (proportional to the mass), *C* = 8*K*^2^*C*_0_/(*Nπ*)^2^ is the motional capacitance (inversely proportional to the stiffness), and *R_m_* = *η_q_/μ_q_C* is the motional resistance related to the dissipative losses. *A* is the electrode area, *ε* and *h* are the dielectric permittivity and thickness of the crystal, respectively; *ω* = *2πf*, where *f* is series resonant frequency, *K* is the electromechanical coupling coefficient, and *N* is integer; *η_q_* is effective viscosity and *μ_q_* is shear stiffness [48].

QCM, QCM-D, and TSM methods are most frequently used in the development of affinity biosensors. However, new opportunities in improved sensitivity are offered by the electromagnetic piezoelectric sensor (EMPAS) developed by Thompson et al. [49]. In this method the shearing oscillations are inducted by a contactless magnetic coil placed under the piezocrystal. The EMPAS can operate up to frequencies close to 1 GHz, which substantially increases the sensitivity of detection.

Another method used in the development of the acoustic affinity sensor is shear horizontal surface acoustic wave (SH-SAW). This method is based on the propagation of an acoustic wave at the surface of the transducer between two interdigital electrodes. The measured shift in resonant frequency and phase is used for evaluation of the mass changes. The sensor operates at frequencies around 100 MHz. The disadvantage of this method is higher sensitivity to the temperature in comparison with the TSM method [50,51].

In the experiments based on the TSM technique, the AT-cut quartz of the fundamental frequency between 5–20 MHz is used. This special cut of the crystal has the advantage of high temperature stability in temperatures around 25 °C—the temperature that is typically used in a biosensing assay [50]. The external high frequency voltage applied to the electrodes at both sides of the quartz crystal transducer induce a shearing wave that propagates perpendicularly to the surface of the transducer toward the aqueous solution (Figure 5A). The mechanical oscillation of the quartz is coupled with electrical values that can be characterized by a Butterworth–Van Dyke equivalent circuit (Figure 5B) [47,52]. The parameters of this circuit can be determined by network analyzer. The typical measuring set-up consists of flow cells with relatively low working volume (below 100 µm). The cells allow connection to the network analyzer or to the QCM instrument [53]. In addition to TSM or QCM techniques, the electrochemical QCM method (EQCM) can also be used. In this case the simultaneous measurement of electrochemical signal and frequency changes offer the advantage of obtaining additional information about the electrical properties of the sensing layer [50,54].

### 2.3. Electrochemical and Acoustic Immunosensors for Leukemia Diagnostics

#### 2.3.1. Electrochemical Cell Immunosensors

The immunosensors for cancer detection can be developed for either cancer markers, for example protein tyrosine kinase 7 (PTK7) that is overexpressed in the membrane of leukemic Jurkat cells, for prostate specific antigen (PSA), or for circulating tumor cells (CTC). In addition, detection of B-cell lymphoma 2 (Bcl-2) and Bcl-2-associated X protein (Bax) can monitor the apoptosis of cancer cells. This allows for an evaluation of the effect of chemotherapeutic drugs. The concentration of these proteins is related to the concentration of the cancer cells [55].

It is also known that oncoviruses induce cancer. For example, bovine leukemia virus (BLV) is a retrovirus that induces enzootic bovine leukosis in cattle [56]. The oncogene infection-induced appearance of specific antibodies can be detected by immunosensors. In paper by Kurtinaitiene et al. [39], the immunosensor in the format presented in Figure 3A was used for detection of antibodies against bovine leukemia protein gp51. Ferrocene carboxylic acid (FCA) and trimethyl hydroquinone, *N*,*N*,*N*′,*N*′-tetramethylbenzidine (TMB) redox markers have been used for amplification of the electron transfer from HRP labelled Abs. In addition, the oncogenes of retroviruses can also be detected by PCR or nucleic acid sensors. For example, in work by Henry et al. [57], the erythroblastic leukemia viral oncogene homolog 2 has been detected by electrochemical genosensor based on immobilization of the DNA probe at the sensor surface and using methylene blue (MB) as a redox marker. MB binds to the guanines in single stranded DNA. In the presence of viral DNA due to increased number of guanines, the redox current measured by differential pulse voltammetry (DPV) increases and serves as a detection signal. Although genosensors are not in the scope of this review, detailed information about this type of electrochemical and acoustic sensor can be found in a review by Lu et al. [58]

Taback et al. [59] reported that cancer markers can also be found in certain concentrations in normal cells of peripheral blood, lymph nodes or bone marrow. Different tumor markers indicate various types of cancers, but one type of cancer can be characterized by more than one kind of tumor marker [60]. Therefore, it is important to develop immunosensors for the detection of several cancer markers. For example, typical cancer markers for breast cancer are human mucin-1 (MUC1), a high molecular weight glycosylated membrane protein produced by epithelial tissues, and carcinoembryonic antigen (CEA)—the membrane glycoprotein involved in cell adhesion. These tumor markers are important for the diagnosis of metastatic breast tumors. In order to avoid false positive results, it is necessary to detect both tumor markers.

As it has been mentioned above, the detection of Bcl-2 and Bax proteins responsible for protection and development of cell apoptosis, respectively, can be rather useful for monitoring the cell apoptosis of cancer cells during treatment with anti-tumor drugs. The level of these proteins can be related to the concentration of tumor cells. This has been shown in paper by Zhou et al. [55], who applied dual electrochemical immunosensors for detection of Bcl-2 and proteins in a suspension of chronic myeloid leukemia (CML) K562 cells. The advantage of dual immunosensors is in more precise detection of the apoptosis by two protein markers. Electrochemical immunosensors has been prepared at the surface of a glassy carbon electrode (GCE) covered by reduced graphene oxide (RGO). The later has good electrical conductivity and a large area for the immobilization of antibodies. The covalent immobilization of antibodies at the surface of RGO is possible due to its carboxylic groups and their activation by NHS/EDC. The antibodies specific to Bcl-2 and Bax were first immobilized on an RGO surface followed by incubation with bovine serum albumin (BSA) to eliminate non-specific interactions. The immunosensor was then immersed into a solution containing a mixture of Bcl-2 and Bax. The detection of these proteins was based on anodic stripping voltammetry (ASV) by using CdSeTe@CdS quantum dots and Ag nanoclusters based on mesoporous SiO_2_ and modified by anti Bcl-2 or anti Bax antibodies, respectively. After formation of the complexes of nanoparticles with the Bcl-2 and Bax proteins at the sensing surface and the release of Cd^2+^ and Ag^+^ ions using HNO_3_, it was possible to detect the proteins using ASV with LOD of 0.5 fmol, which corresponded to approximately 10^3^ cells. The treatment of the K562 cells by the anti-tumor drug nilotinib resulted in a decrease of Bcl-2 (inhibitor of apoptosis) and an increase of Bax concentrations (which promotes apoptosis). The effective concentration of the nilotinib and incubation time was determined. Thus, this interesting approach not only allowed the detection of the CML but also helped in determining its optimal therapy.

Several research studies focused on direct detection of leukemia cells by immunosensors based on immobilization of antibodies on various nanomaterials. Antibodies sensitive to specific proteins overexpressed in the membranes of the leukemia cells were used. For example, in the case of K562 leukemia cells as a cancer marker, P-glycoprotein (P-gp) is often used. In a paper by Du et al. [61], antibodies against P-gp were immobilized at the surface of the AuNPs-methoxysilyl-terminated (Mos) butyrylchitosan modified GCE. After incubation of the sensor with K562 cells, the secondary antibody modified by alkaline phosphatase was added. The cells were detected by measuring the amperometric response of electroactive 1-naphthyl phosphate (see the principle of the detection in Figure 3B). The current was proportional to the logarithm of cell concentrations in the range from 5.0 × 10^4^ to 1.0 × 10^7^ cells/mL, with a detection limit of 1.0 × 10^4^ cells/mL.

Ding et al. [62] proposed direct detection of K562 cells by an impedimetric sensor based on chitosan–AuNPs nanocomposite. The interaction of the cells with the surface of GCE covered by chitosan–AuNPs was performed by electrochemical impedance spectroscopy (EIS) at the presence of 10 mM redox couple K_3_Fe(CN)_6_/K_4_Fe(CN)_6_ (1:1). The adsorption of the cells at the surface resulted in an increase of the charge transfer resistance R_ct_ due to the blocking of the electrode flow from the redox couple. The R_ct_ increased linearly in the logarithmic concentration range of 1.34 × 10^4^ to 1.34 × 10^8^ cells/mL with LOD of 8.71 × 10^2^ cells/mL. However, this assay cannot be considered as an immunosensor because no receptor was used, and the adsorption of the cells was nonspecific. However, in further work, they proposed an EIS immunosensor composed of anti-P-gp mouse monoclonal antibodies immobilized at the surface of GCE covered by an epoxysilane monolayer. This approach allowed detection of K562 cells by measuring R_ct_ values by EIS in the presence of 10 mM redox couple K_3_Fe(CN)_6_/K_4_Fe(CN)_6_ (1:1). The R_ct_ increased linearly in the logarithmic concentration range of 5.0 × 10^4^ to 1.0 × 10^7^ cells/mL, with LOD of 7.1 × 10^3^ cells/mL [63]. A similar EIS cell sensor, but one using GCE covered by AuNPs/polyaniline nanofibers for immobilization of anti P-gp antibodies for detection K562 cells, was reported by Zhang et al. [64]. The R_ct_ increased linearly in the logarithmic range of cell concentrations of 1.6 × 10^2^ to 1.6 × 10^6^ cells/mL, with an LOD of 80 cells/mL.

Rather high sensitivity of detection of K562 cells was demonstrated by Gulati et al. [65] using amperometry and EIS. The anti-P-gp antibodies were immobilized at the single-walled carbon nanotubes (SWCNTs) deposited on a SiO_2_/Si surface. The anodic and cathodic currents were measured by cyclic voltammetry (CV) in the presence of 5 mM redox couple K_3_Fe(CN)_6_/K_4_Fe(CN)_6_ (1:1). The peak current decreased linearly with the increase in the concentration of K562 cells in a logarithmic scale in a range of 1.5 × 10^3^ to 1.5 × 10^7^ cells/mL, with an LOD of 19 cells/mL. Thus, nanomaterials such as SWCNTs are rather useful for the preparation of electrochemical immunosensors, and they provide high sensitivity for leukemia cell detection.

#### 2.3.2. Acoustic Cell Immunosensors

The QCM method has been used for the detection of cancer markers and cells related to leukemia. Despite the simplicity of QCM, so far only very limited papers were published on the detection of cancer markers or cells related to leukemia by mass-sensitive biosensors.

In a paper by Makaraviciute et al. [66], the QCM-D method was used for the detection of bovine leukemia virus antigen gp51. The antibodies specific to gp51 were first dissociated to half antibody fragments (Frag-Ab). Each fragment contained one binding site to the antigen but could be easily immobilized at the gold surface by chemisorption thanks to its sulfuric groups. They showed a higher surface density of antibodies immobilized on AuNPs. This method allowed specific detection of gp51, although the sensitivity and LOD were not reported in this work. The ratio of Δf/ΔD served as a fingerprint for the specific detection of antigens.

The QCM-D method also allowed study of the interaction of the leukemia cells with the endothelial cell system (EC). The EC is important part of the tissues and is responsible for intercellular interactions. It can be activated by various factors, such as tumor necrosis factor α (TNF-α) or leukemia cells, which affect various processes such as inflammation, leukemia, and vascular injuries [67,68]. For example, Pezeshkian et al. [69] demonstrated that acute myeloid leukemia (AML) cells induced activation of the EC with subsequent cell adhesion. In further work from this laboratory, it was shown that by activating the EC using TNF-α it was possible to monitor the interaction of EC with leukemia cells HL-60 and KG-1 by measuring the changes in resonant frequency and dissipation [70].

In a study by Rehman and Zeng [45], the viscoelastic contribution into the interaction of the cells at the surface of a QCM transducer was analyzed. In addition to the frequency changes, Δf, they also measured changes in motional resistance, ΔR_m_, which reflected the viscosity contribution. The slope of |Δf/ΔR_m_| has been used for estimating whether the mass or viscosity effect is dominant. For ideally rigid film, the R_m_ value is practically zero; thus, the frequency changes are proportional to the mass changes. When R_m_ increases, the ratio |Δf/ΔR_m_| decreases and passes through the threshold value below which the changes of the frequency can be attributed to the viscosity effect. For example, the threshold value of |Δf/ΔR_m_| for a 10 MHz crystal is 11.6 Hz/Ω [45]. Thus, above this value the mass effect is dominant. This approach allowed analysis of the binding of monoclonal antibody rituximab to Burkitt’s lymphoma Raji cells. Rituximab is used in cancer immunotherapy. The cells were immobilized at the gold layer of the QCM surface modified by cysteine through tripeptide linker (arginine-glycine-aspartic acid). The tripeptide was attached to the cysteine by glutaraldehyde. It has been demonstrated that the addition of cells to the surface of the QCM transducer resulted in a decrease of frequency and in an increase of motional resistance. A similar tendency was observed for interaction of rituximab with the cells. From the changes of frequency and motional resistance, they determined the value |Δf/ΔR_m_| = 24 Hz/Ω for binding the cells to the QCM surface. This was higher than the threshold value. This means that the QCM response is mainly due to the mass effect. The binding of rituximab to the Raji cells resulted in |Δf/ΔR_m_| = 14.8 Hz/Ω, which was also above the threshold value. Because the mass effect was dominant, they used the Sauerbrey equation and Langmuir isotherm for determination of the apparent binding constant of rituximab to the Raji cells, k = 1.6 × 10^6^ M^−1^. Thus, the QCM method is rather useful for the study of the cell-surface and antibody–cell interactions.

Unfortunately, the QCM method has not been used yet for the detection of leukemia cells by antibodies immobilized at the surface. There is only one report by Hao et al. [51] that applied a shear horizontal surface-acoustic-wave (SH-SAW) method for the detection and separation of Jurkat and K562 leukemic cells. The CD3 antibodies were immobilized at the silanized SAW surface. The center frequency was 122.5 MHz. The maximal sensitivity of detection was 10^3^ Jurkat cells/mL, with detection time of 15 min. Using this sensor, it was possible to separate the cells in a mixture of Jurkat/K562 cells = 1:1000 at a concentration of the cells of 10^6^ cells/mL.

Table 1 summarizes the properties of electrochemical and acoustic immunosensors for the detection of leukemia cells.

## 3. Nucleic Acid Aptamers in Diagnostics of Leukemia

### 3.1. Nucleic Acid Aptamers

Nucleic acid aptamers were discovered 30 years ago independently by three laboratories in the USA, together with the in vitro method of their selection—the SELEX (systematic evolution of ligands by exponential enrichment) [14,73]. They are also known as artificial antibodies, which is due to their high affinity to the target that is comparable to that of monoclonal antibodies. However, they have only been used as receptors in biosensors since 2000 [74]. The interest in aptamers for application in biosensors as well as in medicine is tremendous and growing rapidly. Other than for their use as biosensors, the aptamers have been examined for use as drugs in the therapy as well as for the recognition of elements for targeted drug delivery by polymer nanoparticles [75].

The term “aptamer” originates from the Latin *aptus*, “*to fit*”, and the Greek meros, “*the part*”. Aptamers are single stranded DNA or RNA oligonucleotides typically composed of 15–80 bases. In a solution, they fold into a 3D structure forming a binding site to the corresponding target. As a target, a practically unlimited number of compounds can be used for aptamer selection, such as low molecular species and proteins but also viruses, bacteria, or cells and tissues. In contrast with antibodies, the aptamers are more stable. Once the aptamer sequence is selected, it can be synthesized by standard nucleic acid chemistry methods. Both ends of aptamers can be chemically modified by linkers for their immobilization at the surfaces (thiol, amino, biotin) or by optical or redox probes that allow detection of the binding with the target. The chemically modified aptamers prevent the degradation by endonucleases in biological samples. In contrast with antibodies, nucleic acid aptamers are flexible molecules. They can form molecular beacons. In addition, using simple molecular engineering based on the hybridization of complementary parts, the aptamers mono or heterodimers can be prepared. These dimers (aptabodies) are characterized by two binding sites and by a higher affinity to their target [76]. The application of aptamers in medicine is also advantageous because they do not induce any significant immune response. Due to their low molecular weight (~8–25 kDa), aptamers can penetrate the tissue faster than antibodies [77]. Aptamer-based sensors can be regenerated using soft conditions (2M NaCl, 0.2 M glycine, heating) that allow dissociation of the aptamer–target complex. After regeneration, the aptasensor can be reused several times. This is important for the development of reusable biosensors. There are several types of interactions between aptamer and ligand, such as hydrogen bonds, hydrophobic and electrostatic interactions, van der Waals interactions, or aromatic stacking [78,79]. The dissociation constant for the aptamer–target complex is typically in the pico–nanomolar range [77].

RNA aptamers are smaller than DNA aptamers with the same number of nucleotides. After chemical modification, RNA aptamers are better suited for the transport of substances into cells because they more easily cross the membrane. However, they are less stable in comparison with DNA aptamers [80]. The number of articles on the application of aptamers in various fields of biomedical research has increased substantially. Special focus is on the application of aptamers in diagnostics and therapy of cancer.

Up to now, aptamers have already been developed for various targets, such as thrombin [81], HBV virus [82], *E. Coli* [83] and other bacteria, aflatoxin [84,85], prostate specific antigen (PSA) [86], CEA tumor markers [87], or cell membrane proteins such as protein tyrosine kinase 7 (PTK7) in the CCRF-CEM T-lymphoblast membrane and others [88]. In the case of cancer, the aptamers can recognize oncoproteins, cancer markers, and metabolites associated with cancer processes in the body [89].

In recent years, great effort was made in the selection of aptamers for cancer cells, especially for their membrane proteins. For this purpose, Cell-SELEX has been shown as more suitable in comparison with traditional SELEX, which is appropriate for the development of aptamers for water soluble molecules. Cell-SELEX is important for the development of aptamers for tumor markers (Figure 6). This is a method that selects aptamers specific to the cancer cell membrane proteins in their natural environment. Cell-SELEX has the advantage of working without a requirement for separate purified proteins [90,91].

Cell-SELEX starts similarly to SELEX by generating a random library of DNA sequences. This is followed by their incubation with target cells and the removal of unbound sequences. Bound sequences are eluted from the cell membrane proteins by heating to 95 °C. An important step is incubation of the obtained sequences with negative cells, thereby removing the sequences with lower affinity. The resulting aptamers are amplified by polymerase chain reaction (PCR). If more specific aptamers are needed, the Cell-SELEX process can be repeated several times. Each subsequent generation of aptamers has better binding properties in comparison with previous generations [92].

### 3.2. Aptamers for Cancer Markers

Detection of tumor markers is expensive and time consuming by currently used clinical methods. Therefore, substantial effort is focused on the development of biosensors based on antibodies and nucleic acid aptamers.

In 2008, Shangguan et al. [88] reported a new strategy for the detection of tumor markers using Cell-SELEX. By applying this method to leukemia cells, they selected an sgc8c DNA aptamer that bound to CCRF-CEM cells, T-lymphoblasts of acute lymphoblastic leukemia (ALL). This aptamer recognized with high affinity the protein tyrosine kinase 7 (PTK7), which was confirmed by both gel electrophoresis and flow cytometry. Aptamers for various cancer cells and cancer markers have been selected by a similar method. The most often used aptamers that recognize leukemia cells are listed in Table 2. Other aptamer sequences used for the detection leukemia cells or as cancer markers, as well as those used in the therapy, can be found in a recent review by Nur et al. [93].

### 3.3. Electrochemical and Acoustic Aptasensors

#### 3.3.1. Electrochemical Aptasensors for Detection Leukemia Cells

Electrochemical methods of detection are very efficient, especially for aptasensors. In addition to electrochemical impedance spectroscopy (EIS), other techniques such as differential pulse voltammetry (DVP) or square wave voltammetry (SWV) can also be applied for detection of the interaction between the redox labelled aptamers and analytes of interest, including cells. The methods of immobilization of the aptamers at the solid support are like those for antibodies (Figure 2). However, aptamers can be much more easily immobilized by maintaining proper orientation of their binding sites [74]. In addition, the flexibility of nucleic acid aptamers in the formation of aptamer beacons or aptamer dimers improves the sensor design toward higher sensitivity [76]. The advantageous aptamer conformation is a molecular beacon in which the terminal parts of the aptamer are composed of a short complementary chain of approximately 4–5 base pairs. One aptamer end is modified by a redox label, such as ferrocene (Fc) or methylene blue (MB), while other—supporting part—is modified by a thiol, amino, or biotin group allowing aptamer mobilization at solid support. The binding of the cell to the aptamer results in dissociation of the hydrogen bonds that stabilize the double strand, promote formation of the binding site, and as a result, the redox label is moved away from the sensing surface. This causes a decrease of the redox current that serves as an analytical signal. Figure 7 summarizes various designs of electrochemical aptasensors for cell detection. The label-free method (Figure 7A) uses a redox couple in a buffer, for example, 1 mM Fe(CN)_6_^3−/4−^. Incubation of the sensor with the cells reduces charge transfer from the redox label to the sensor surface, which results in an increase of the charge transfer resistance (EIS method) or a decrease of the current (amperometry). In an aptamer beacon configuration, one nucleotide chain is modified by the redox label (ferrocene or methylene blue). In the presence of cells, the hydrogen bonds dissociate, and the redox label is moved away from the sensing surface (signal off) (Figure 7B). Figure 7C illustrates the signal on configuration. Without cells, the aptamers are in random conformation. The addition of the cells results in the formation of the binding site. The redox label approaching the sensor surface, and the current increases. Another variation of a signal-on configuration is presented in Figure 7D. The aptamer is hybridized with a short complementary chain. The addition of the cells results in the dissociation of the duplex, and the redox label moves toward the sensor surface, which causes an increase of the current.

Practically all of the aptasensor designs shown in Figure 7 have been used in the detection of cells. Pan et al. [18] were among the first to report an aptasensor based on DNA aptamer (sgc8c) specific to PTK7 receptors in leukemia cells. The aptamers modified at 5′-end by the thiol group were chemisorbed at the gold surface. The EIS method in the presence of 10 mM redox couple Fe(CN)_6_^3−/4−^ has been used for the detection of CCFR-CEM leukemia cells. The incubation of the sensor with cells resulted in an increase of the charge transfer resistance, R_ct_, with increasing cell concentration. A linear dependence of R_ct_ vs. logarithm of the cell concentration was observed in the range of 1 × 10^4^–1 × 10^7^ cells/mL with the LOD = 6 × 10^3^ cells/mL. The response of the sensor was specific. Insignificant changes of R_ct_ values occurred in the presence of control Raji cells. However, the relatively low sensitivity of this EIS sensor was not enough for clinical applications. A similar approach was used in further research by Babelova et al. [102,103], in which Jurkat leukemia was used as a target, and U266 control cells were used. The LOD for Jurkat cell detection was 105 ± 10 cells/mL. A small non-specific interaction was observed for the U266 control cells. In order to study the possible reason for this non-specificity, small unilamellar liposomes with a diameter of approximately 20 nm and composed of a 1:1 (mol/mol) mixture of 1,2-dimyristoyl-sn-glycero-3-phosphocholine (DMPC) and 1,2 dimyristoyl-sn-glycero-3-glycerol (DMPG) were used. The sensor response was like those for the U266 cells. It was proposed that certain nonspecific interactions can be due to interference of the aptamers with the lipid part of cells.

An exceptionally high sensitivity of CCFR-CEM leukemia cell detection (10 cells/mL) was reported by Chen et al. [104] using the multivalent detection principle. They took advantage of glycan expression at the surface of CCFR-CEM cells for their immobilization at the surface covered by Concanavalin A (Con A). Con A is lectin that binds to a broad spectrum of saccharides but presents a good platform for immobilization of cells at the sensing surface. Con A has been immobilized at the surface of GCE covered by a conducting layer composed of graphene oxide (GO) and poly(amidoamine) (PAMAM) dendrimers. The electrochemical detection was performed by incubation of the cells with gold nanoparticles (AuNPs) modified by HRP and thiolated sgc8c aptamers specific to PTK7. While aptamers provided specific binding of AuNPs to the cell surface, the HRP in the presence of the substrate—H_2_0_2_ and hydroquinone (HQ)—resulted in the generation of the electrical signal. The detection of leukemia cell was specific, as revealed in experiments using several control cell-lines.

The sandwich principle was used by Yu et al. [97] for the detection of the human chronic myelogenous leukemia (CML) K562 cell line by electrochemical aptasensor with high sensitivity (LOD = 79 cells/mL). They applied DNA aptamers (T2-KK1B10) sensitive to K562 cells developed by Sefah et al. [96] but extended by 10 thymines at the 5′-end to provide better flexibility of the aptamers (see Table 2). The thiolated aptamers were chemisorbed at a gold electrode. After incubation with the K562 cells, the biotinylated Con A was added. A strong binding to the mannose that covered the cell membrane occurred. Then, alkaline phosphatase conjugated with streptavidin (ST-ALP) was added. The electroactive product of degradation of α-naphtyl—the 1-naphtol—was detected by DPV (see Figure 3B for the principle of the detection). The sensor was selective, and other cell lines did not cause a significant response even at 10-fold higher concentrations. The aptasensor was regenerated by rinsing with double distilled water at 70 °C. After 5 cycles of regeneration, the sensor retained 89.6% of its initial current response. The cell sensor was validated in spiked blood plasma with good recovery in the range of 79.6% to 93.3%, depending on the cell concentration.

A high-sensitive aptasensor for electrochemical detection of CCRF-CEM cells (LOD 10 cells/mL) was proposed by Khoshfetrat and Mehrgardi [105]. The thiolated sgc8c aptamers in a hairpin configuration were immobilized on magnetic Fe_3_O_4_ nanoparticles coated by gold. The double stranded hairpin part of the aptamer allowed intercalation of the redox probe ethidium bromide (EB). In the presence of leukemia cells, the aptamers lost their hairpin structure, EB was released, and the current decreased. The detection was performed by the DPV method on a screen-printed electrode modified by nitrogen-doped graphene that provided higher conductivity in comparison with original graphene. The sensor could differentiate between leukemia cells and Ramos control cells.

The labelling of the aptamers by the redox probes offers the advantage of a one-step detection of cancer cells with good sensitivity. This approach has been used for the detection of Jurkat cells by sgc8c aptamers modified at the 3′ end by ferrocene (Fc) or methylene blue (MB). The aptamers were immobilized on the gold surface by chemisorption or biotin-neutravidin methods [103,106]. The redox potential for these probes is different. While for MB the redox potential is around –0.34 V, while for Fc it is 0.4 vs. the Ag/AgCl reference electrode. For the cell–aptamer interaction, the signal-off principle was used (See Figure 7B). In this approach the aptamers were in the molecular beacon configuration. The redox label was close to the electrode surface and the well resolved current peak was measured by the DPV method. The addition of the cells resulted in disruption of the hydrogen bonds that stabilized the aptamer beacon, and the redox label moved away from the electrode surface. This resulted in a decrease of the current peak amplitude. For optimization of the sensor properties, it is also important to select the proper aptamer concentration for their immobilization at surface. Lower concentrations cannot provide enough sensitivity due to the low surface density of the aptamers. However, too high a surface density of aptamers can reduce aptamer flexibility and proper folding upon the binding of the cells. It is also important to select an optimal incubation time for the sensing surface with the cells. However, it should be noted, that during preparation of the calibration curve, the current arising from the redox probes should be stable. Therefore, it is crucial to check whether the current peak is stable in time during potential cycling (using cyclic voltammetry (CV)). The examples of Jurkat cell detection by aptasensors modified by MB or Fc probes using DPV methods are shown in Figure 8. Figure 8A shows the DPV for the aptasensor based on the MB label. At the potential around –0.34 V vs. Ag/AgCl, MB is reduced to leucomethylene blue by 2 electrons [107]. Therefore, at the reduction potential, the maximum current is observed. Incubation of the aptamers with Jurkat cells resulted in a decrease of this current (see Figure 7 and text above for explanation). A similar response was also obtained for Fc-labeled aptamers.

Based on DPV, it is possible to construct calibration curves for cell detection by MB and Fc-modified aptamers. This is shown in Figure 8B. A rapid increase of the relative changes of the peak current at low concentration of the cells can be seen. The limits of detection for MB- and Fc-labelled aptasensors were similar: 38 ± 8 cells/mL and 37 ± 6 cells/mL, respectively. Much smaller changes of the current were observed for control U266 cells that did not contain PTK7 receptors. Thus, the sensors based on redox-labeled aptamers provided rather high sensitivity and good selectivity of Jurkat cells. As it can be seen from Table 3, the detection limits obtained by MB- and Fc-labeled aptasensors are among the lowest in comparison with other electrochemical and acoustic aptasensors for cell detection.

An interesting assay for diagnosis of acute lymphoblastic leukemia (ALL) was proposed by Mazloum-Ardakani et al. [122]. They combined a genosensor detecting the BCR-ABL1 mutant gene and an aptasensor detecting the CEA cancer marker. The presence of Philadelphia chromosome (BCR-ABL1)-positive (Ph^+^) is a high risk for leukemia development in children. Only 20–30% of children with Ph^+^ and ALL survived [123]. Carcinoembryonic antigens (CEA, CD66 family) were also considered as important cancer markers for early detection of ALL [124]. In this approach the DNA genosensor was used first for evaluation of whether the BCR-ABL1 wsa present. A positive test indicated that the patient already had ALL or was at high risk to be affected by ALL. Subsequently, the second test performed by an aptasensor sensitive to CEA in blood could indicate the presence of ALL. For the development of the DNA sensor and aptasensors, the authors used nanocomposite materials consisting of carbon quantum dots (C-dots) and AuNPs. Both C-dots and AuNPs were immobilized at the surface of GCE. The thiolated ssDNA or aptamer probes were chemisorbed at the surface of the AuNPs. Both NPs enhanced the electrochemical signal. A DPV method was used for the detection of DNA hybridization with ssDNA probe as well as for CEA marker detection by DNA aptamers in the presence of electroactive probe—2 mM catechol. In both cases—DNA hybridization and binding of CEA to the aptamer—the current decreased, indicating the presence of either the BCR-ABL1 gene or CEA in the blood. The electrochemical sensors allowed detection of the BCR-ABL1 gene and CEA with a LOD of 1.5 pM and 0.26 pg/mL, respectively. The sensors were validated using blood from patients with various stage of ALL.

A diagnosis of leukemia is possible not only by direct detection of leukemia cells but can also be based on the detection of certain biomarkers. Among them, lysozyme (Lys) and interferon gamma (IFN-γ) have been considered to be very useful. Detection of Lys can be beneficial in distinguishing various leukemia types [125,126], while concentration of IFN-γ indicates the cellular immune level [127]. For diagnostic purposes, it is important to simultaneously detect both Lys and IFN-γ. This was performed by Xia et al. [128], who reported a dual electrochemical biosensor based on aptamers that selectively bound those two markers. One aptamer for Lys detection was modified by methylene blue (MB), while other for IFN-γ was modified by ferrocene (Fc). Both aptamers were associated with partially complementary chains. The sensor design and signaling mechanisms were like those presented in Figure 7D. However, the signal-off principle was held for the detection of IFN-γ, and the signal-on principle was used for the detection of Lys. The SWV method allowed simultaneous detection of IFN-γ and Lys by measurement of the amplitude of the current originating from the redox probes of Fc and MB. Both peaks were well separated. While peak current for the Fc-labelled strain is maximum at 275 mV, those modified by MB are maximum at –292 mV vs. the saturated calomel reference electrode (SCE). The LOD of 1.14 pM and 16.4 pM were obtained for simultaneous detection of IFN-γ and Lys, respectively. Better sensitivity for IFN-γ could be due to the lower dissociation constant of the corresponding aptamers. The sensor was specific and validated in human serum. The advantage of the sensor was its reusability after up to five regeneration cycles.

The electrochemical aptasensor for Lys detection was also reported in a paper by Jarczewska et al. [129]. In contrast with a previous probe, they used free MB in a buffer. Due to the positive charge of this redox indicator, a strong electrostatic attraction took place with the DNA bases (especially guanines). The thiolated aptamers specific to Lys were immobilized at the gold layer of the working electrode, incubated with MB. Using SWV, the amplitude of the cathodic current peak was measured before and after the addition of Lys. Prior to Lys addition, the aptamers were in random conformation. The addition of Lys resulted in aptamer folding due to the formation of the binding site. MB probes moved closer to the sensing surface, and the cathodic current peak increased. A rather high sensitivity of Lys detection was achieved; LOD = 0.1 fM in a concentration range 0.1 fM to 1 nM. The sensor was verified in 1% human serum spiked with 10 nM of Lys with a recovery of 84%.

#### 3.3.2. The Combined Antibodies and Aptamer-Based Assay

In paper by Li et al. [60], a combined aptamer- and antibody-based biosensor was reported for simultaneous detection of MUC-1 and CEA markers. The thiolated DNA aptamers were chemisorbed at the surface of a gold electrode. Incubation with MCF-7 breast cancer cells resulted in the formation of a stable complex with aptamers at the sensor surface. The addition of anti-CEA antibodies modified by CdS nanoparticles were further used for identification of the CEA tumor marker at the surface of the MCF-7 cells. After an acid dissolution step using HNO_3_, the Cd^2+^ ions released from the complex were measured by anodic stripping voltammetry at the surface of a glassy carbon electrode coated by mercury film. A well-resolved voltammetric peak was measured at –0.708 V vs. saturated calomel electrode (SCE). It was shown that a significant peak was observed only if both MUC1 and CEA markers were present. The sensor was specific, and no peaks occurred when acute leukemia CCRF-CEM cells were examined instead of MCF-7 cells. The minimum detectable concentration of MCF-7 cells was 3.3 × 10^2^ cell/mL, with a wide linear range of 10^4^–10^7^ cell/mL. This method can also be applicable for the detection of leukemia cells. In this case, the aptamer selective to the PTK7 cancer marker and anti-PTK7 antibodies can be used. Both PTK7 specific aptamers and antibodies are available but have not been used yet for leukemia cell detection in the sandwich format described above.

Several examples of electrochemical cell sensors based on DNA aptamers were reviewed in papers by Justino et al. [12], Yazdian-Robati [130], and Nur et al. [93]. Recent papers by Suhito et al. [131] reviewed achievements in development of both immuno- and aptasensors for cell detection.

#### 3.3.3. Acoustic Aptasensor for Detection of Leukemia Cells

The advantage of acoustic methods is their label-free detection of cells. The aptamers are immobilized at the surface of a piezoelectric transducer, and the main measuring signal is the resonant frequency of the crystal oscillation. Most often, the thickness shear mode (TSM) method is used, which also allows evaluation of the motional resistance, R_m_, which reflects the viscoelastic contribution. The TSM method in its simple set up is known as a quartz crystal microbalance (QCM) (see Section 2.2.2. for details). However, there are also certain difficulties in the evaluation of the response of the binding of the cells to the sensor surface. When cells specifically interact with the sensing surface, it typically causes a decrease of the resonant frequency, which is a result of the increase of the mass or thickness of the sensing layer. At the same time, due to shearing oscillation, the interaction of the surface with surrounding aqueous solution takes place. The surface viscosity can also affect the resonant frequency changes. In addition, the penetration depth of the acoustic wave (approximately 0.25 µm) is lower than the thickness of the cell layer at the piezocrystal surface (approximately 3 µm for Jurkat and MOLT-4 cells (see corresponding AFM images published in [121])). Therefore, the sensor only partially “feels” like added mass. The QCM method has so far been used mostly for the detection of drugs [132], DNA [133], peptides [134], proteins [135,136], viruses, or bacteria [137] and to a lesser extent for the detection of cancer cells [59,138]. However, this method has shown that it can be useful for diagnostic purposes.

So far, only a very limited number of papers have been published with a focus on the detection of leukemia cells. Among the first was a paper by Pan et al. [119] in which the detection of CCRF-CEM cells containing PTK7 cancer markers was made using sgc8c aptamers conjugated with magnetic beads. The magnetic beads allowed better separation of the cells with adsorbed aptamers. The LOD of leukemia cells based on measuring the decrease of the resonant frequency was 8 × 10^3^ cells/mL. Better sensitivity (LOD = 1160 cells/mL) of detection of CCRF-CEM cells was achieved by a QCM aptasensor with amplified detection using the silver enhancement method [120]. In this work, sgc8c aptamers were chemisorbed at the gold layer of the QCM transducer. After incubation of the sensor with the CCRF-CEM cancer cell line, the gold nanoparticles modified by p-aminophenylboronic acid (APBA-AuNPs) were added at the sensing surface. The APBA at the surface of the AuNPs formed the complex, with sialic acid in a glycan structure covering the cells. At the same time, the AuNPs served as a catalyst for reduction of silver ions in the presence of hydroquinone.

Babelova et al. [102] used the TSM method for the detection of leukemia Jurkat cells with the LOD of 463 ± 50 cells/mL using the sgc8c aptamers chemisorbed at the gold layer of quartz crystal. This approach was also used in a paper by Poturnayova et al. [121] in which in addition to chemisorption of the aptamers the biotin-neutravidin method was also used for aptamer immobilization.

In this work, Jurkat and MOLT-4 leukemia cells were used. The latter may slightly differ by concentration of PTK7 receptors. Figure 9A shows the scheme of the piezocrystal surface covered with aptamers and adsorbed cells. In addition to the resonant frequency the motional resistance, R_m_, was also determined. While resonant frequency decreased with an increasing concentration of leukemia cells, the R_m_ value increased (Figure 9B). The decrease of the resonant frequency is evidence of adsorption of the cells at the sensing layer. This effect is most probably due to the changes in the mass of the sensing layer. The increase of the motional resistance is evidence of viscosity contribution. The most remarkable changes were observed for aptasensors based on chemisorbed aptamers. The changes in the frequency (Δf = –9 Hz) were already significant at a concentration of 50 MOLT-4 cells/mL. The changes of resonant frequency and motional resistance at the maximal examined concentration of the cells (5 × 10^5^ cells/mL) was higher for MOLT-4 cells (Δf = −121.2 ± 12.0 Hz and ΔR_m_ = 6.9 ± 1.1 Ω) than for Jurkat cells (Δf = −97.6 ± 9.4 Hz and ΔR_m_ = 2.5 ± 0.8 Ω). This can be probably due to the higher concentration of PTK7 receptors at the membranes of MOLT-4 cells. The addition of control U266 cells did not cause significant changes of resonant frequency and motional resistance. This is evidence of good specificity of the cell sensor.

Using a similar approach as in a paper by Rehman and Zeng [45], it is possible to estimate the viscosity contribution to the mass changes from the ratio |Δf/ΔR_m_|. For an 8 MHz crystal, the threshold value |Δf/ΔR_m_| = 10.37 Hz/Ω. Above this value, the frequency changes are mainly due to changes of mass, while below the threshold value the viscosity contribution is dominant. At the highest concentration of MOLT-4 cells (5 × 10^5^ cells/mL) |Δf/ΔR_m_| = 17.6 Hz/Ω and those for Jurkat cells is 39.04 Hz/Ω. These values are significantly higher than the threshold value. This means that the changes of frequency are related mainly to the changes of the mass. The maximal sensitivity of detection of leukemia cells corresponded to LOD = 195 ± 20 cells/mL. The sensor was regenerable after incubation in 0.2 M glycine solution for 10 min and subsequent washing by phosphate buffer saline (PBS). After regeneration, the sensor was reusable at least three times without significant loss of sensitivity. This is a great advantage over immunosensors for which the application of some regenerating agents such as glycine resulted in loss of binding affinity. The sensor was validated in spiked diluted blood plasma with recovery of approximately 95%.

Thus, the acoustic sensors offer the advantage of label-free detection of cells and easy evaluation of the sensor response based on the changes of resonant frequency. Simultaneous detection of motional resistance by the TSM method allows estimation of viscosity contribution. It seems that the changes of the frequency due to the interaction of the cells with aptamers immobilized at the piezocrystal are mostly due to the mass changes. As it can be seen from Table 1 and Table 3, the sensitivity of acoustic immuno- and aptasensors is lower in comparison with most sensitive electrochemical sensors. However, acoustic sensors, in the case of additional amplification using nanoparticles, can be more advantageous for application in complex biological samples due to insensitivity to possible interferences with redox species that can affect the signal in electrochemical detection. As it can be also seen from comparison of Table 1 with Table 3, the sensitivity of immunosensors and acoustic sensors is comparable. However, as we already mentioned, the aptasensors are more advantageous due to higher stability, lower cost, better flexibility of aptamers, easier aptamer immobilization, chemical modification, and finally due to the possibility of regeneration for multiple use.

## 4. Conclusions

This review clearly demonstrates an increasing interest in the development of biosensors for the diagnosis of leukemia. Most straightforward is the detection of cancer cells using antibodies or nucleic acid aptamers specific to certain cancer markers, such as PTK7 proteins in the membranes of leukemia cells. However, detection using other markers that accompany the development of cancer disease is also useful for the detection of oncogenes. A successful prognosis in cancer therapy requires early diagnosis and thus high sensitivity of cancer detection in complex biological samples such as blood or blood plasma. Most sensitive immunosensors and aptasensors that can detect cancer cells with sensitivity below 10–50 cells/mL are suitable for this purpose, considering that in order to avoid the matrix effect it is necessary to perform analysis in diluted samples. As we have shown, the immuno- and aptasensors are of comparable sensitivity. This conclusion is based on a comparison of the results focused on separately developed immuno- and aptasensors under conditions that are not fully identical. The most correct analysis requires a comparison of the properties of these two sensor types prepared under similar conditions and preferably in one laboratory. This has not been done yet for leukemia cell detection. Increased interest in the development of aptasensors also reflects their advantage over immunosensors. Among them, one can mention higher stability, flexibility, easier chemical modification, and immobilization at surfaces as well as the possibility of sensor regeneration. However, there is still an absence of commercialization of the biosensors that have been developed so far. Only available are ELISA kits for some cancer markers such as prostate specific antigen (PSA), human cancer antigen 125 (CA125), or β2-Microglobulin. ELISA kits that are suitable for the detection of cancer cells are not yet available in the market. Considering the rather high stability of DNA aptamers, future commercialization of these cell sensors is very likely. However, substantial effort is needed for the development of aptamers and their chemical modification to increase their ionic strength, to make them less sensitive to temperature, and to reduce interferences with other species in complex biological liquids.

## Figures and Tables

**Figure 1 biosensors-11-00177-f001:**
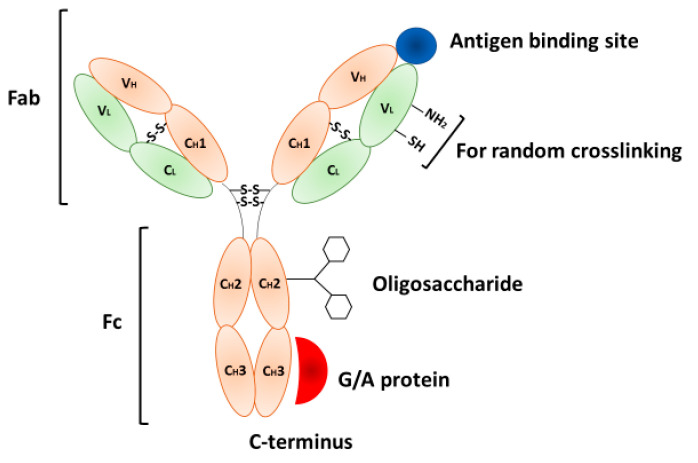
The scheme of the structure of the antibody. G/A proteins specifically interact with the Fc part. -NH2 and -SH groups are used for random covalent immobilization of Abs. Adopted from Li and Chen [26].

**Figure 2 biosensors-11-00177-f002:**
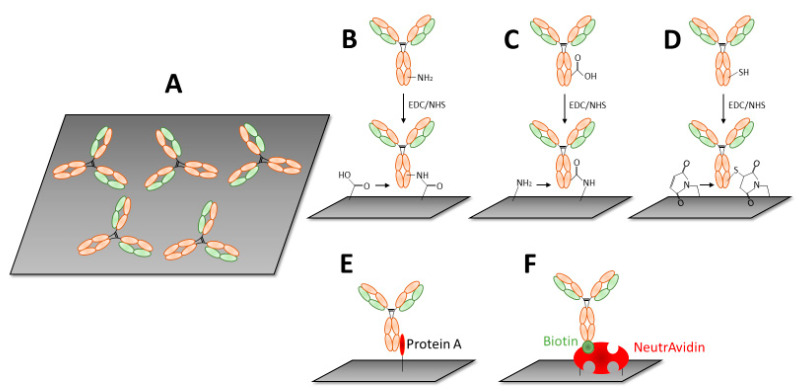
The scheme of the most frequently used methods of immobilization of antibodies at surfaces. (**A**) Physical adsorption; (**B**) covalent immobilization to 11-mercaptoundecanoic acid (MUA); cysteamine (**C**), or maleimide (**D**). (**E**) Protein A-Abs complexes; (**F**) biotinylated Abs immobilized at the surface of neutravidin monolayer. Partially adopted from Li and Chen 2018 [26].

**Figure 3 biosensors-11-00177-f003:**
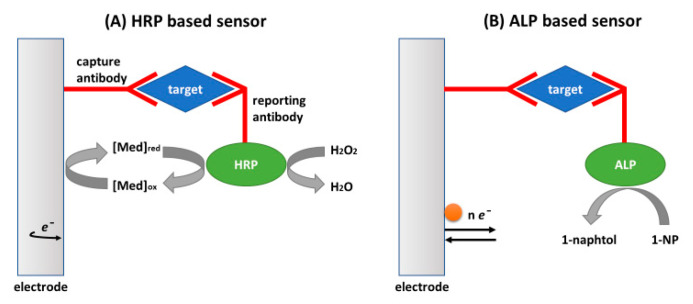
The scheme of the design of enzyme-based immunosensors that involve (**A**) horseradish peroxidase (HRP) and (**B**) alkaline phosphatase (ALP). Partially adopted with from [36]. 2016 Elsevier.

**Figure 4 biosensors-11-00177-f004:**
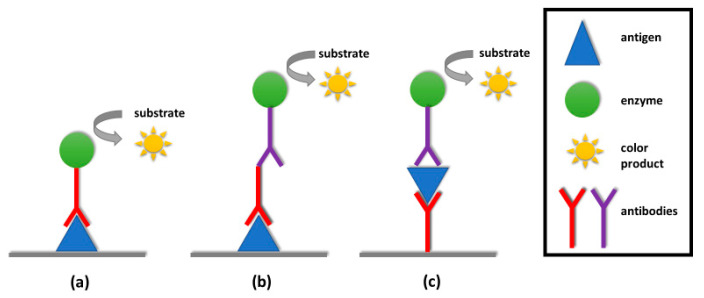
The scheme of ELISA: (**a**) direct assay; (**b**) indirect assay using enzyme modified Ab that selectively binds to a certain part of primary Ab; (**c**) indirect assay based on interaction of secondary antibody modified by enzyme with another binding site at the antigen. Adopted from [41].

**Figure 5 biosensors-11-00177-f005:**
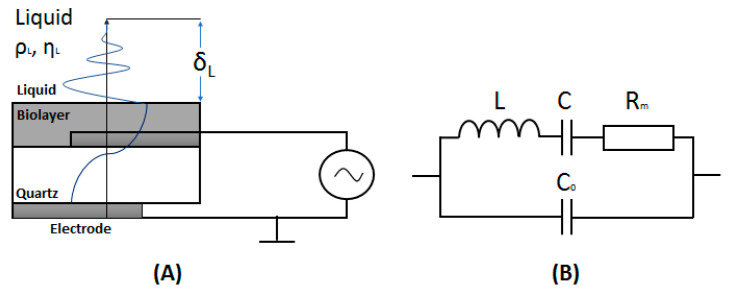
(**A**) Scheme of the propagation of the acoustic wave. *η_L_* and *ρ_L_* are the viscosity and density of liquid, respectively. δ is penetration depth. (**B**) Butterworth–van Dyke (BvD) equivalent circuit [45].

**Figure 6 biosensors-11-00177-f006:**
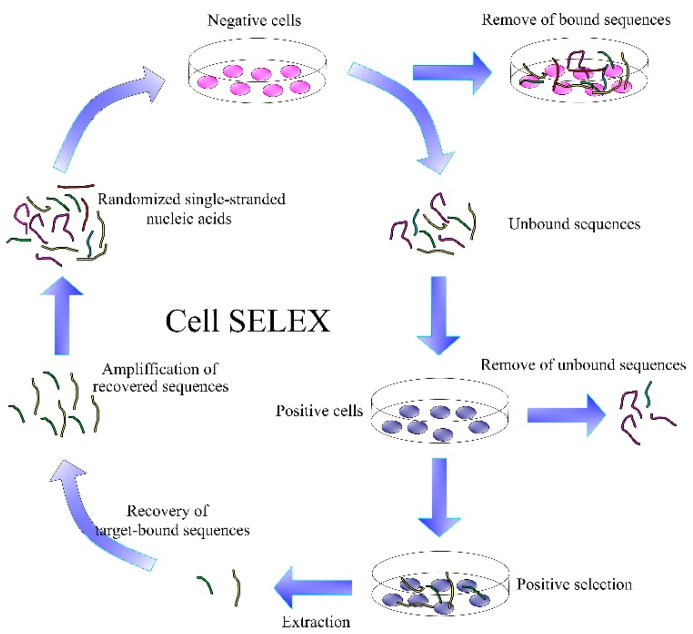
The scheme of Cell-SELEX adapted from [75].

**Figure 7 biosensors-11-00177-f007:**
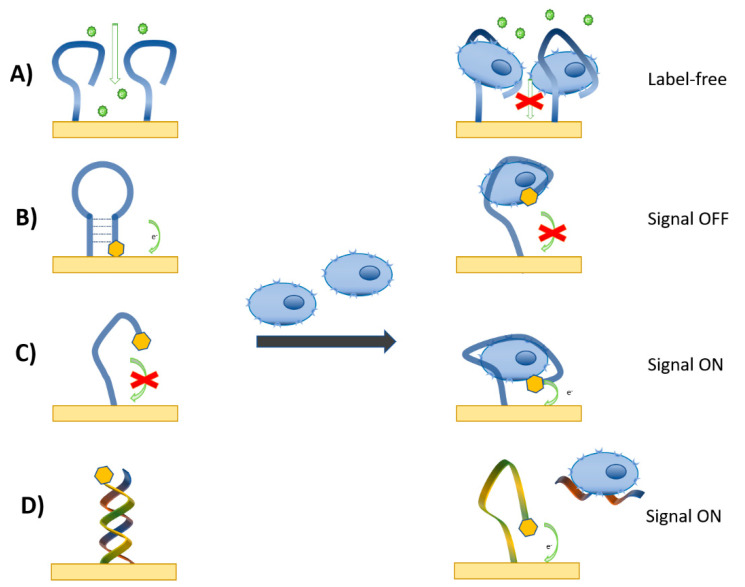
The scheme of various designs of the electrochemical aptasensors for the detection of cells. (**A**) Label-free detection in the presence of the redox couple in a buffer. (**B**) Aptamer beacon. (**C**) Signal-on configuration. (**D**) Signal-on configuration using aptamer hybridized with complementary chain. For explanation, see the text above.

**Figure 8 biosensors-11-00177-f008:**
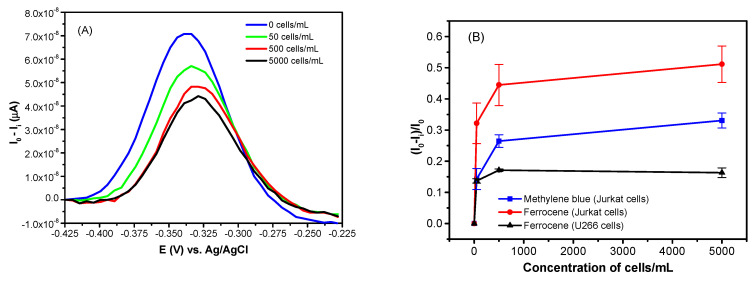
(**A**) Differential pulse voltammograms of methylene blue modified aptamer at different Jurkat cell concentrations in working buffer at pH 7.4. (**B**) The relative changes of the peak current intensity of the two types of electrochemical aptasensors vs. cell concentration. I_0_ is peak current prior to addition and I_i_ after addition of the cells in respective concentrations. Partially adapted from [106].

**Figure 9 biosensors-11-00177-f009:**
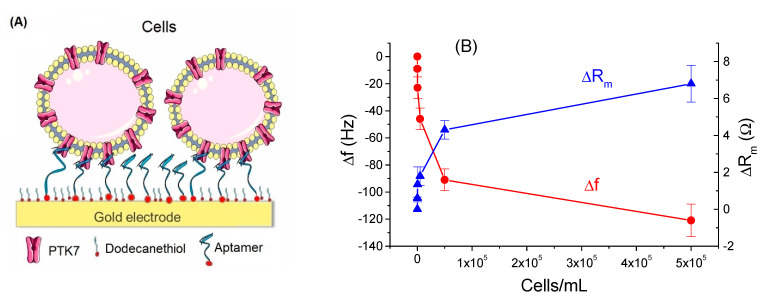
(**A**) The scheme of the sensing surface. Thiolated sgc8c aptamers are chemisorbed at the gold surface. 1-dodecanethiol formed hydrophobic patches at uncovered parts of the gold surface. (**B**) Plot of the changes of the resonant frequency, Δf, and motional resistance, ΔR_m_, vs. concentration of MOLT-4 leukemic cells. Partially adopted with permission from [121] 2018 Wiley.

**Table 1 biosensors-11-00177-t001:** The properies of the electrochemical and acoustic immunosensors for the detection of leukemic cells.

Cell	Surface for Antibody Immobilization	Method of Detection	Linear Range, Cells/mL	LOD, Cells/mL	Reference
K562	AuNPs/butyrylchitosan/GCE	Amperometry ALP-modified secondary antibody	5 × 10^4^–10^7^	1.0 × 10^4^	[61]
K562	epoxysilan/GCE	EIS	5 × 10^4^–10^7^	7.1 × 10^3^	[63]
K562	AuNPs/polyaniline nanofibers/GCE	EIS	1.6 × 10^2^–1.6 × 10^6^	80	[64]
K562	SWCNTs/SiO_2_/Si	CV	1.5 × 10^3^–1.5 × 10^7^	19	[65]
K562	MWCNTs	EIS	2 × 10^3^–2 × 10^6^	11	[71]
KG1a	GCE/GQDs/AuNPs	Amperometry	1–25	1	[72]
Jurkat	SiO_2_	SH-SAW	-	10^3^	[51]

ALP, alkaline phosphatase; AuNPs, gold nanoparticles; CV, cyclic voltammetry; EIS, electrochemical impedance spectroscopy; GCE, glassy carbon electrode; GQDs, graphene quantum dots; MWCNTs, multiwalled carbon nanotubes; SH-SAW, shear horizontal-surface acoustic; SWCNTS, single-walled carbon nanotubes.

**Table 2 biosensors-11-00177-t002:** Aptamers to various types of leukemia.

Disease	Target Protein	Aptamer	Aptamer Sequence 5′–3′	K_D_, nM	Reference
Acute lymphoblastic leukemia	PTK7 CCRF-CEM cells)	sgc8c	ATC TAA CTG CTG CGC CGC CGG GAA AAT ACT GTA CGG TTA GA	0.78	[18]
Chronic lymphocytic leukemia	Chemokine ligand CXCL12 (MS-5 cells)	NOX-A12	NH_2_-(CH_2_)_6_-OP(O)(OH)O-GCG UGG UGU GAU CUA GAU GUA UUG GCU GAU CCU AGU CAG GUA CGC	0.20	[94,95]
Chronic myelogenous leukemia	K562 cell	T2-KK1B10	TTT TTT TTT TAC AGC AGA TCA GTC TAT CTT CTC CTG ATG GGT TCC TAT TTA TAG GTG AAG CTG T	-	[96,97]
Acute myeloid leukemia, breast cancer	Nucleolin (MV4-11 cells, MCF-7 cells)	AS1411	GGT GGT GGT GGT TGT GGT GGT GGT GG	*-*	[98]
Acute myeloid leukemia	Unknown protein (HL60 cells)	KH1C12	ATC CAG AGT GAC GCA GCA TGC CCT AGT TAC TAC TAC TCT TTT TAG CAA ACG CCC TCG CTT TGG ACA CGG TGG CTT AGT	4.5 ± 1.6	[96]
Acute myeloid leukemia	Sigles-5 (NB4 cells)	K19	AAG GGG TTG GGT GGG TTT ATA CAA ATT AAT TAA TAT TGT ATG GTA TAT TT	12.37	[99]
Burkitt’s lymphoma	Immunoglobulin heavy chain of IgM (Ramos cells)	TD05	ACC GGG AGG ATA GTT CGG TGG CTG TTC AGG GTC TCC TCC CGG TG	7.9–359	[100,101]

**Table 3 biosensors-11-00177-t003:** Comparison of the properties of electrochemical and acoustic aptasensors for detection cancer cells.

Cell Line	Aptamer	Immobilization Method	Method	Linear Range Cells/mL	LOD Cells/mL	Ref.
CCRF-CEM	sgc8c	Self assembly of thiol-terminated aptamer	EIS, CV	1 × 10^4^–1 × 10^7^	6 × 10^3^	[18]
CCRF-CEM	sgc8c	GCE/rGO	DPV	10^2^–5 × 10^4^	10	[104]
CCRF-CEM	sgc8c	SPCE/Fe_3_O_4_ mganetic nanoparticles coated by Au	DPV	10–10^6^	10	[105]
CCRF-CEM	sgc8c	MWCNTs	DPV	10–5 × 10^5^	8	[108]
CCRF-CEM	sgc8c	PAA	CV	10^2^–10^6^	100	[109]
CCRF-CEM	sgc8c	Graphene/AuNPs/Fe_3_O_4_	SWV	5–500	3	[110]
DLD-1	MUC-1-aptamer	MUC-1 aptamer bound on CNSs	EIS, CV	1.25 × 10^2^–1.25 × 10^6^	40	[111]
CT26	SBA-15-prNH2	AuNPs	EIS, CV	10–10^5^ (CV), 10^5^–6 × 10^6^ (EIS)	2	[112]
HepG2, HeLA	HeLa-aptamer	SCPE-NH_2_-modified aptamer	EIS	-	163.7	[113]
HL-60	KH1C12	GCE/PDCNs/DSNPs	CV	10^3^–10^6^	660	[114]
HL-60	KH1C12	Paper-based microporous support doped by Au	DPV	5 × 10^2^–7.5 × 10^7^	350	[115]
HL-60	KH1C12	GCE/AuNPs/poly(3,4-ethylenedioxythiophene)	EIS	25–5 × 10^5^	250	[116]
K562	T2-KK1B10	Fe_3_O_4_ nanoparticles	DPV	14–14 × 10^5^	14	[117]
K562	T2-KK1B10	Au electrode/sandwich assay	DPV	10^2^–10^7^	79	[97]
K562	T2-KK1B10	GCE/GO	ASV	10^2^–10^7^	60	[118]
Jurkat	sgc8c	Self assembly of thiol-terminated aptamer + MCH	EIS	50–500 × 10^3^	105 ± 10	[102]
Jurkat	sgc8c	Chemisorption of thiolated sgc8c modified by MB	DPV	50–500	38 ± 8	[106]
Jurkat	sgc8c	Biotinylated aptamers modified by Fc at Au surface	DPV	50–500	37 ± 6	[106]
CCRF-CEM	sgc8c	Aptamers conjugated to the magnetic beads	Magnet QCM	10^4^–1.5 × 10^5^	8 × 10^3^	[119]
CCRF-CEM	sgc8c	Self assembly of thiol-terminated aptamer + MCH	QCM	2 × 10^3^–1 × 10^5^	1160	[120]
Jurkat	sgc8c	Self assembly of thiol-terminated aptamer + dodecanethiol	QCM	50–500 × 10^3^	463 ± 50	[102]
MOLT-4	sgc8c	Self assembly of thiol-terminated aptamer + dodecanethiol	TSM	5–500	195 ± 20	[121]

ASV, anodic stripping voltammetry; AuNPs, gold nanoparticles; CNSs, carbon nanospheres; CV, cyclic voltammetry; DPV, differential pulse voltammetry; EIS, electrochemical impedance spectroscopy; Fc, ferrocene; GCE, glassy carbon electrode; GO, graphene oxide; MB, methylene blue; MCH, 11-mercaptohexanol; MWCNTs, multiwalled carbon nanotubes; PAA, porous anodic alumina; PEDOT, poly(3,4-ethylenedioxythiophen; rGO, reduced graphene oxide; SCPE, screen-printed carbon electrode; SWV, square wave voltammetry; QCM, quartz crystal microbalance; TSM, thickness shear mode.

## Data Availability

Not applicable.

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
