# Peer review of "Advances in Electrochemical and Acoustic Aptamer-Based Biosensors and Immunosensors in Diagnostics of Leukemia"

_biosensors, 2021, doi:10.3390/bios11060177_

Round 1
Reviewer 1 Report
The paper reports advances in electrochemical and acoustic aptamer-based biosensors and immunosensors in the diagnostics of leukemia. In general, the manuscript is well-organized, logically laid out. This paper is possibly publishable but should be revised again. For improving a manuscript, it is advisable to address the following comments:
- In Section 2.1, it is recommended to add one useful reference (https://doi.org/10.1016/j.bios.2019.03.024), which immobilized the antibody on the particular nanostructured surface by the covalent bond.
 - In Section 2.1, more details can be provided to discuss in terms of proteins A and G for Abs immobilization.
 - The overall writing of the manuscript could be improved. For example, “Electrochemical and Acoustics Methods of the Detection Ab-Ag interactions” on line 241 of page 6, could be revised to “Electrochemical and Acoustics Methods for the Detection of Ab-Ag Interactions”.
 - More references could be added in Table 1, and the current references are up to date.
 - The resolution of Figure 5 is too low to read, please change a better one.
 - Do you need to put the LOD in Table 2?

Author Response
We are grateful to this reviewer for careful reading of the manuscript and for most useful comments that allowed us to improve manuscript.
Comment 1: In Section 2.1, it is recommended to add one useful reference (https://doi.org/10.1016/j.bios.2019.03.024), which immobilized the antibody on the particular nanostructured surface by the covalent bond.
Response: The reference has been included in the revised manuscript and short description of the immobilization method has been added in the Section 2.1.
Comment 2: In Section 2.1, more details can be provided to discuss in terms of proteins A and G for Abs immobilization.
Response: We extended the part dealing with A and G proteins and included additional reference.
Comment 3: The overall writing of the manuscript could be improved. For example, “Electrochemical and Acoustics Methods of the Detection Ab-Ag interactions” on line 241 of page 6, could be revised to “Electrochemical and Acoustics Methods for the Detection of Ab-Ag Interactions”.
Response: The manuscript has been improved as suggested by reviewer.
Comment 4: More references could be added in Table 1, and the current references are up to date.
Response: The Table 1 has been updated by new references as requested. We also updated Table 3 and additional references were included.
Comment 5: The resolution of Figure 5 is too low to read, please change a better one.
Response: The resolution of Figure 5 was improved.
Comment 6: Do you need to put the LOD in Table 2?
Response: Table 2 shows only the sequences of DNA aptamers. The binding properties are characterized by constant of dissociation, KD. LOD is not relevant for this table, because it depends on the concrete sensor design and detection method used.
Reviewer 2 Report
The paper needs a major revisit for the English language editing and primarily for the structuring of sentences. I recommend reaching out to professional English language editing services at the university. The content is very interesting and of great relevance to the field of biosensors and the audience of the journal. It is well structured and provides sufficient detail of the two types of sensors selected- immuno sensors and acoustic based sensors. See the attached file for some revisions of English language. It is not comprehensive and needs a second pair of eyes to completely look presentable.

Author Response
Comment: The paper needs a major revisit for the English language editing and primarily for the structuring of sentences. I recommend reaching out to professional English language editing services at the university. The content is very interesting and of great relevance to the field of biosensors and the audience of the journal. It is well structured and provides sufficient detail of the two types of sensors selected- immuno sensors and acoustic based sensors. See the attached file for some revisions of English language. It is not comprehensive and needs a second pair of eyes to completely look presentable.
Response: We are grateful to this reviewer for careful reading of the manuscript and for most useful comments in enclosed pdf file. All comments raised by reviewer were properly addressed. In particularly, new reference related to the droplet digital PCR (ddPCR) has been included. Figure 5 and 7 legends were reduced, and the detailed description has been moved to the main text. The language revision was also made.
Reviewer 3 Report
This review article by Tibor Hianik introduces an area of diagnostic aptamers that has remained largely unstudied. Particularly in the field of hematological neoplasia, sensitive methods for the detection of minimal residual disease are required, which can be used to complement the current PCR analyses. Among other things, the time factor plays an essential role. The sooner the result of the MRD analysis is available , the sooner therapeutic intervention can be made if molecular recurrence is detected .
The procedures covered and discussed in the review article point here to an important improvement in current diagnostic procedures. 
Comments:
1. the article is too long and should be shortened in essential parts.
2. the introduction introduces the nomenclature of leukemias. This part of the manuscript should be critically read and modified by a specialist in the field. The who classification was mentioned by the author. This should be elaborated in the context.
3. line 909. in a review article, own work should be cited neutrally.
Author Response
We are grateful to this reviewer for careful reading of the manuscript and for most useful comments that allowed us to improve manuscript.
Comment 1: the article is too long and should be shortened in essential parts.
Response: The article has been shortened in some parts. However, due to necessity of updating Tables 1 and 3 we included additional references.
Comment 2: the introduction introduces the nomenclature of leukemias. This part of the manuscript should be critically read and modified by a specialist in the field. The who classification was mentioned by the author. This should be elaborated in the context.
Response: The nomenclature has been discussed with clinical specialist as recommended. This part was also shortened. We kept only most necessary information.
Comment 3: line 909. in a review article, own work should be cited neutrally.
Response: We agree with this comment and corrected the text correspondingly.